# Platelet Inhibition by Low-Dose Acetylsalicylic Acid Reduces Neuroinflammation in an Animal Model of Multiple Sclerosis

**DOI:** 10.3390/ijms22189915

**Published:** 2021-09-14

**Authors:** Anna Vogelsang, Susann Eichler, Niklas Huntemann, Lars Masanneck, Hannes Böhnlein, Lisa Schüngel, Alice Willison, Karin Loser, Bernhard Nieswandt, Beate E. Kehrel, Alexander Zarbock, Kerstin Göbel, Sven G. Meuth

**Affiliations:** 1Department of Neurology with Institute of Translational Neurology, University Hospital Münster, 48149 Münster, Germany; eichlers@uni-muenster.de (S.E.); niklas.huntemann@uni-muenster.de (N.H.); lars.masanneck@uni-muenster.de (L.M.); hannes.boehnlein@web.de (H.B.); kerstin.goebel@ukmuenster.de (K.G.); 2Department of Neurology, University Hospital Düsseldorf, 40225 Düsseldorf, Germany; meuth@uni-duesseldorf.de; 3Department of Anesthesiology, Intensive Care and Pain Medicine, University Hospital Münster, 48149 Münster, Germany; lisa.schuengel@uni-muenster.de (L.S.); kehrel@uni-muenster.de (B.E.K.); zarbock@uni-muenster.de (A.Z.); 4The Northern Foundation School, Newcastle-upon-Tyne University Hospitals, Newcastle-upon-Tyne NE15 8NY, UK; alice.willison@newcastle.ac.uk; 5Department of Human Medicine, Institute of Immunology, Carl von Ossietzky University Oldenburg, 26129 Oldenburg, Germany; karin.loser@uol.de; 6Rudolf Virchow Center, Research Center for Experimental Biomedicine, University of Würzburg, 97080 Würzburg, Germany; bernhard.nieswandt@virchow.uni-wuerzburg.de

**Keywords:** acetylsalicylic acid, experimental autoimmune encephalomyelitis, platelets, multiple sclerosis, thromboxane, glycoprotein VI, platelet factor 4

## Abstract

Aside from the established immune-mediated etiology of multiple sclerosis (MS), compelling evidence implicates platelets as important players in disease pathogenesis. Specifically, numerous studies have highlighted that activated platelets promote the central nervous system (CNS)-directed adaptive immune response early in the disease course. Platelets, therefore, present a novel opportunity for modulating the neuroinflammatory process that characterizes MS. We hypothesized that the well-known antiplatelet agent acetylsalicylic acid (ASA) could inhibit neuroinflammation by affecting platelets if applied at low-dose and investigated its effect during experimental autoimmune encephalomyelitis (EAE) as a model to study MS. We found that oral administration of low-dose ASA alleviates symptoms of EAE accompanied by reduced inflammatory infiltrates and less extensive demyelination. Remarkably, the percentage of CNS-infiltrated CD4^+^ T cells, the major drivers of neuroinflammation, was decreased to 40.98 ± 3.28% in ASA-treated mice compared to 56.11 ± 1.46% in control animals at the disease maximum as revealed by flow cytometry. More interestingly, plasma levels of thromboxane A_2_ were decreased, while concentrations of platelet factor 4 and glycoprotein VI were not affected by low-dose ASA treatment. Overall, we demonstrate that low-dose ASA could ameliorate the platelet-dependent neuroinflammatory response in vivo, thus indicating a potential treatment approach for MS.

## 1. Introduction

Multiple sclerosis (MS) is the most common immune-mediated inflammatory disease of the central nervous system (CNS) [1,2]. It is generally accepted that an aberrant immune response results in demyelination and axonal loss within the CNS, leading to neurological disability [3]. Although the etiology of MS is not fully understood, advances in our understanding of the immune mechanisms that characterize the relapsing and inflammatory stages of MS have led to therapeutic options with distinct immunomodulatory or immunosuppressive effects in the past 30 years. Thus, the currently available disease-modifying drugs for treating MS mainly target lymphocytes, including consequences on T cell activation, as well as on peripheral T lymphocyte trafficking into the CNS [4,5]. While various treatments reduce the frequency and severity of relapses, they cannot cure the disease, emphasizing that other, not typically considered, components of the immune system might also be involved. There is evidence that interaction between platelets and immunity fosters aberrant immune responses in MS and experimental autoimmune encephalomyelitis (EAE), an animal model that mimics pathological hallmarks observed in MS [6]. In EAE, platelets were found accumulating in the CNS parenchyma, thereby amplifying the (neuro)inflammatory response. As already demonstrated for cells of the immune system, platelets, as part of the coagulation system, seem to infiltrate the CNS and aggregate within MS and EAE lesions [6,7,8,9,10,11]. Of note, platelet depletion during the effector phase of EAE reduces the disease severity and is associated with a reduction in CNS-infiltrating immune cells from the periphery [6,10]. Furthermore, recent studies demonstrated that platelets are chronically activated in the early stages of EAE [8]. Platelets obtained from MS patients, especially in relapsing-remitting and secondary progressive MS, show a chronically activated status [12,13,14]. Interestingly, MS patients showed an increased risk of vascular disorders, such as ischemic stroke and myocardial infarction [15,16,17]. The critical role of platelets in MS and EAE pathophysiology still needs to be clarified in more detail.

One of the most widely used antiplatelet medications is the anti-inflammatory drug acetylsalicylic acid (ASA; also known as aspirin). ASA has many uses, one of which is to diminish platelet activation and aggregation, thereby limiting thrombus formation [18]. The mechanism of action of ASA involves irreversible acetylation and subsequent inactivation of cyclooxygenase (COX)-1, thus inhibiting the formation of thromboxane A_2_ (TxA_2_), a key factor in platelet aggregation during primary hemostasis [19]. At low doses, ASA can be taken life-long by patients and is, in contrast to high-doses, effective by selective inhibition of platelet aggregation [20], making it a useful instrument to prevent occlusive vascular events, such as ischemic stroke or myocardial infarction [21]. High doses of ASA do not exhibit an increased antithrombotic effect compared to low doses of ASA and, due to the risk of bleeding, high-dose ASA is only used for short-term analgesia in clinical practice [22,23]. Of note, high-dose ASA also results in the inhibition of endogenous prostacyclin biosynthesis [24]. Due to the irreversible inhibition of COX-1, the therapeutic effect of ASA is dependent on platelet lifespan and synthesis. Further, ASA affects the coagulation cascade and fibrinolysis by reducing thrombin levels at sites of microvascular injury and by impairing fibrin clotting [25,26]. Of particular relevance, ASA-induced inhibition of TxA_2_ synthesis systemically blocks platelet cluster of differentiation (CD)40L release [27,28], a pro-thrombotic and proinflammatory protein that was shown to be upregulated in MS patients [29,30,31].

Few studies have examined the effect of ASA on EAE disease activity. Early data, published in 1949, indicated that high-dose prophylactic treatment with sodium salicylate, the precursor of ASA, reduced EAE disease severity [32]. In an EAE model using Lewis rats, treatment with sodium salicylate delayed disease onset and reduced clinical signs of neuroinflammation [33]. In addition, Mondal et al. demonstrate that ASA impedes EAE disease progression protecting regulatory T cells and suppressing proinflammatory T helper (T_H_) 1 and T_H_17 cells in a relapsing-remitting EAE model [34]. More recent studies have examined the role of COX-1 and COX-2 in EAE. Naproxen, an inhibitor of COX-1 and -2, was shown to delay the onset of EAE symptoms and alleviate disease severity [35]. Here, we investigated the therapeutic potential of low-dose ASA under pathophysiological conditions in vivo.

## 2. Results

### 2.1. ASA Significantly Ameliorates Clinical Symptoms of EAE

To elucidate the role of ASA during neuroinflammation, mice subjected to EAE were orally treated daily with 3 mg/kg body weight (BW) ASA (ASA_low_), 30 mg/kg BW ASA (ASA_high_) or a vehicle control for a period of 30 days, beginning on the day of immunization. Interestingly, prophylactic treatment with both high- and low-dose ASA significantly alleviated disease severity and cumulative (cum.) EAE scores but no difference could be detected between low and high doses of ASA (Figure 1B,C). Disease onset was not affected by ASA treatment (Figure 1D). Disease incidence was the same for all groups, with all mice used in experimentation showing EAE symptoms (data not shown). Histological examination of sections of the lumbar spinal cord, which is the main site of neuroinflammatory processes in active EAE [36], showed reduced CNS infiltration, as well as less demyelination (Figure 1E–H). However, there was no significant difference in the number of activated microglia and macrophages visualized by ionized calcium-binding adaptor molecule 1 (Iba-1) between all experimental groups (Appendix A). Single stainings are given for all fluorochromes (Appendix A). 

### 2.2. ASA Reduces the Infiltration of CD4^+^ T Lymphocytes into the CNS 

Further investigations using flow cytometry showed a reduction of peripheral infiltrates within the CNS of both ASA-treated mice, as indicated by amelioration of invaded CD45^high^ cells (Figure 2B). The total number of lymphocytes was also reduced in both ASA-treated mice, while monocytes/macrophages, dendritic cells and microglia were not significantly different (Figure 2C). Since EAE lesions are dominated by infiltrated proinflammatory T lymphocytes [36], we investigated the proportion of lymphocyte cells within the inflammatory infiltrates. Indeed, the percentages of CD4^+^ T cells, CD8^+^ T cells, and also B cells were significantly reduced in mice treated with a low or high dose of ASA (Figure 2D,E). The proportion of infiltrated CD4^+^ T cells, for instance, was reduced to 40.98 ± 3.28% in mice treated with low-dose ASA compared to 56.11 ± 1.46% in control mice (Figure 2D). Since the depletion of platelets during the effector phase is associated with an improved EAE course [6], we determined the number of platelets present in the CNS of EAE affected mice. Indeed, we did not find significant differences in the platelet counts within the inflamed CNS at the disease maximum between the experimental groups. Interestingly, no significant difference between low- or high-dose administration of ASA was observed in terms of cum. EAE scores, the degree of demyelination, the total number of CNS infiltrating leukocytes, lymphocytes, CD4^+^ T cells, CD8^+^ T cells and B cells.

### 2.3. TxA_2_ Synthesis Is Reduced by Low-Dose ASA Treatment

To establish the underlying mechanism of low-dose ASA-induced platelet-mediated inhibition of neuroinflammation, soluble factors were measured using enzyme-linked immunosorbent assays in plasma of naïve untreated as well as immunized EAE mice treated with low-dose ASA or vehicle. The production of thromboxane B_2_, the stable hydration product and, therefore, the representative for TxA_2_ levels, by platelets, was significantly lowered in mice receiving low-dose ASA compared to vehicle-treated EAE mice (Figure 3A). Since platelet factor 4 (PF4) is a well-known proinflammatory platelet alpha-granule protein, we investigated plasma PF4 level changes during neuroinflammation. However, in agreement with a former study, we found no effect of ASA treatment on PF4 plasma levels [37]. GPVI is one of the most important platelet collagen receptors, and soluble GPVI is rapidly proteolytically cleaved by TxA_2_-induced matrix metalloproteinases (MMP) [38]. However, no significant differences between ASA-treated or control mice were found for soluble PF4 or GPVI plasma levels (Figure 3B,C). Interestingly, soluble GPVI levels were significantly increased in EAE mice at the disease maximum (Figure 3C), implying a potential contribution of GPVI to EAE pathology. To this end, we investigated the GPVI-related effects on neuroinflammation in EAE. The clinical EAE scores showed a minor reduction in mice deficient for GPVI (GPVI^-/-^) (Appendix A). The cum. EAE scores between WT and GPVI^-/-^ mice were not altered (Appendix A). Neither disease incidence nor onset differed between the experimental groups (Appendix A).

### 2.4. Therapeutic Treatment with Low-Dose ASA Improves the Clinical Outcome of EAE

ASA was also administered therapeutically to mice and was started on the day of disease onset. Clinical EAE scores and cum. EAE scores were reduced by low-dose ASA treatment (Figure 4A,B). Since the ASA treatment was started on disease onset, the incidence and day of disease onset were displayed as an additional methodological control factor. In agreement, the disease incidence was not altered and was 100% as all mice developed EAE symptoms (Figure 4C). The disease onset was also not affected by treatment with low-dose ASA (Figure 4D).

These data suggest that the application of low-dose ASA is a promising approach to abrogate the neuroinflammatory processes mediated by activated platelets responsible for MS pathogenesis. Low-dose ASA is of particular interest due to its low bleeding risk and well-tolerated side effects, but with a comparable impact as high-dose ASA on neuroinflammation.

## 3. Discussion

Using established therapies for new indications, instead of searching for novel therapeutic approaches for disease pathologies that are not yet fully understood, is known as “repurposing”. This method is becoming increasingly popular in translational science, as it has the clear advantage of side effects already well-studied in human clinical trials, thereby increasing patient safety and accelerating the approval process for new indications.

In the present study, we demonstrate that the pharmacologic blockade of platelets using both low- and high-dose ASA, a well-known antiplatelet drug, renders EAE mice less susceptible to neuroinflammation. Oral treatment with ASA was associated with a reduction in disease severity, a lessened degree of demyelination and decreased immune cell infiltration into the CNS. These findings indicate that there are meaningful relationships between low-dose ASA-induced platelet inhibition and amelioration of disease pathogenesis. We tested two doses of ASA and found that low-dose ASA resulted in an equal reduction in EAE progression, cum. EAE scores, the extent of demyelination and CNS infiltration by immune cells compared to the high-dose ASA treatment. Consequently, low-dose ASA has an equal anti-inflammatory effect on the progression of neuroinflammation while having a much better side effect profile. One explanation for this could be that the inactivation of COX-1 by ASA is irreversible, and even low-dose ASA can reach an inactivation saturation phase [19]. 

The effect of low-dose ASA cannot be explained by a deficiency of infiltrated platelets because flow cytometric analysis showed no significant difference in the number of infiltrated platelets in the CNS of ASA-treated mice compared with control mice. We suggest that the beneficial effect of ASA treatment during EAE is mediated by altered platelet function. In line with the results from our chronic EAE model, Mondal et al. demonstrated a promising effect of low-dose ASA in a relapsing-remitting EAE model. In addition, Modal et al. investigated the EAE course in a chronic EAE model and found the same trend supported by our data. However, they focused on EAE course progression and performed experiments on the relapsing-remitting model [34]. The contribution of platelets has to be investigated in a relapsing-remitting model of EAE and is not well understood.

Examination of the mechanism of platelet inhibition by ASA demonstrated a significant reduction of TxA_2_ and recruitment of CD4^+^ T cells in ASA-treated mice. Furthermore, TxA_2_ may mediate the interaction between platelet and lymphocyte or endothelial cell activation. Interestingly, Kabashima et al. have shown in mice that naïve T cells express the thromboxane receptor and can be activated by TxA_2_ [39]. One hypothetic mechanism would be that TxA_2_ directly activate CD4^+^ T cells, while ASA is able to reduce this activation. However, the thromboxane and prostacyclin systems also regulate the interactions of platelets with endothelial cells in the vessel wall, and ASA is known to repeal endothelial dysfunction [40,41,42,43]. Endothelial cells are able to produce prostacyclin, an effective vasodilator and inhibitor of platelet activation as well as aggregation [44,45,46], whereas platelets synthesize TxA_2_, a vasoconstrictor and platelet aggregator [47,48,49]. There is also some evidence that endothelial cells are able to produce TxA_2_, although this is still controversial [50,51,52]. Therefore, another hypothesis of possible mechanisms of the reduced recruitment of CD4^+^ T cells would be altered activation of endothelial cells by TxA_2_ and ASA.

The development of EAE is dependent on the timing of platelet depletion [6], and not only prophylactic but also therapeutic treatment with low-dose ASA demonstrated a reduction in EAE symptoms. In addition, it is widely accepted that autoreactive T cells generated in the periphery migrate across the BBB, inducing disseminated inflammatory lesions within the brain parenchyma, leading to demyelination. Therefore, a prophylactic treatment might influence this early lymphocytic activation in contrast to a therapeutic application. Subsequently, prophylactic ASA treatment may thereby reduce immune cell infiltration into the CNS, explaining the more pronounced amelioration of the EAE clinical course compared to therapeutic application.

Since PF4 is a proinflammatory platelet alpha-granule protein and plasma levels of PF4 are significantly increased in MS patients compared to healthy control [37], we investigated plasma PF4 level changes during neuroinflammation. However, no difference in PF4 levels was found between ASA- and vehicle-treated mice or between naive and EAE mice. In agreement with our data, Cananzi et al. demonstrated PF4 levels were not affected by ASA treatment in MS patients [37]. GPVI is one of the most essential platelet collagen receptors involved in atherothrombosis, and soluble GPVI is rapidly proteolytically cleaved by TxA_2_-induced MMP [38,53]. Since GPVI expression is restricted to platelets and megakaryocytes, it is another potential antithrombotic target [54,55], and in line, patients with a GPVI deficiency do not exhibit symptoms other than a mild bleeding tendency [56,57,58,59,60,61,62,63]. In the present study, the reduction of soluble GPVI plasma levels was not found in low-dose ASA-treated mice compared to control mice.

Interestingly, a promising difference was found between naïve and EAE mice: Soluble GPVI plasma levels were increased during the EAE disease course compared to naïve mice, suggesting an important role of GPVI for platelets during neuroinflammation. In contrast to high-dose ASA, GPVI antagonistic agents do not lead to increased bleeding risk, making GPVI, in addition to low-dose ASA, a promising target in antiplatelet therapy [64]. However, we detected only a minor amelioration of EAE symptoms in GPVI deficient mice. Kleinschnitz et al. addressed the involvement of GPIb and GPVI in experimental stroke and found that the blockade of these receptors results in a reduction of the infarct size, with the effect induced by a GPVI blockade not as pronounced as the inhibition of GPIb. In addition, anti-GPVI-treated mice showed a better neurological outcome. However, the difference did not reach statistical significance [53]. Targeting GPVI is clearly a less compelling treatment strategy for autoimmune neuroinflammation, as it is a less effective abrogator of the neuroinflammatory response. Since GPVI acts as a platelet activation receptor, its function might be compensated by alternative pathways, such as G-protein-mediated signaling pathways [65]. A comparative study discovered that a combined approach, including both GPVI inhibition and ASA treatment, leads to an additional decrease of plaque-induced platelet aggregation in vitro [66]. Therefore, combination therapy of GPVI blockade and ASA could be a promising therapeutic approach to investigate in EAE. However, one should keep in mind that TxA_2_ is required for hemostasis in GPVI-deficient mice, which could cause significantly prolonged bleeding times [65]. 

The reduction in inflammatory infiltrates in ASA-treated mice raises the question if the interaction between platelets and lymphocytes is direct or indirect. The interaction between platelets and lymphocytes might lead to the production of signaling molecule production between these cells. In the present study, we focused on the platelet-mediated impact on neuroinflammation, but further experiments could address the complex interaction of platelets and lymphocytes by measuring, for example, differences in CD40L, CD62P or first procaspase activating compound-1 levels, as well as platelet-lymphocyte aggregates following ASA treatment [8,67].

In summary, additional studies are needed to further reveal the underlying mechanisms of platelet inhibition by low-dose ASA during neuroinflammatory processes. Overall, our findings show that the inhibition of platelet activation via low-dose ASA is beneficial in an animal model of MS, indicating a potential antithrombotic strategy to combat MS.

## 4. Materials and Methods

### 4.1. Study Approval

All animal experiments were conducted in accordance with the local animal ethics committee in North Rhine-Westphalia, Germany (81-02.04.2018.A382, approval date: 8 February 2019; 84-02.04.2013.A142, approval date: 16 October 2013) and the ARRIVE (Animal Research: Reporting of In Vivo Experiments) guidelines [68].

### 4.2. Mice 

C57BL/6J mice were obtained from Charles River Laboratories (Sulzfeld, Germany), and GPVI^-/-^ were generated as previously described [69]. Mice were kept in individually ventilated cages under standard conditions. Male and female, 8–15-week-old mice were used for immunization, and all animals had access to food as well as water ad libitum. 

### 4.3. EAE Induction

For active EAE induction, mice were immunized with 200 µg MOG_35–55_ peptide (peptide sequence MEVGWYRSPFSRVVHLYRNGK; Charité-University Hospital Berlin, Germany) dissolved in 200 µL complete Freund’s adjuvant (Merck KGaA, Darmstadt, Germany). Complete Freund’s adjuvant including 200 µg *Mycobacterium tuberculosis* (strain H37 Rα; Becton, Dickinson and Company (BD), Sparks, United States of America (USA)) and 100 µL MOG_35-55_ emulsion were administered into each flank proximal to the inguinal lymph nodes by subcutaneous injection under isoflurane anesthesia. On the day of immunization and two days later, 100 ng pertussis toxin (lot number 1007 with potency 1.5; Hooke Laboratories Inc., Lawrence, KS, USA) diluted in 100 μL Dulbecco’s phosphate-buffered saline (DPBS; Merck KGaA) was injected intraperitoneally. Clinical symptoms of EAE were assessed daily in a blinded fashion by two independent observers and translated into clinical EAE scores: score 0: no detectable clinical signs; score 1: partially limp tail; score 2: paralyzed tail; score 3: moderate hindlimb weakness; score 4: complete hindlimb weakness; score 5: mild paraparesis of the hindlimbs; score 6: paraparesis, weakness in forelimbs; score 7: severe paraparesis or paraplegia; score 8: tetraparesis; score 9: quadriplegia or premoribund state; score 10: death. Mice with a score of ≥1 for two consecutive days were included in the study. Animals with a score > 7, with a score of 7 for more than 4 consecutive days or with a weight loss of more than 20% of the initial body weight were excluded from the experiment to guarantee animal welfare, and their last observed score was assumed for the rest of the experiment. The cumulative score was calculated as the average of all daily scores for the whole period of the experiment. 

### 4.4. Treatment with ASA

Mice were treated with 3 mg/kg BW ASA (ASA_low_; Merck KGaA), 30 mg/kg BW ASA (ASA_high_) or vehicle prophylactically (beginning at the day of immunization) or therapeutically (beginning at the day of disease onset (day 10)) once daily. ASA was dissolved in 0.5% carboxymethyl cellulose (CMC; Merck KGaA), and mice were gavaged with 500 µL of ASA solution. Control mice were gavaged with 500 µL 0.5% CMC as vehicle. 

### 4.5. Immunohistochemistry

After intensive transcardial perfusion of mice with DPBS at disease maximum (day 16), lumbar spinal cords were dissected in 10 µm-thin slices using a cryotome (Leica, Wetzlar, Germany). Sections were stained with hematoxylin and eosin (H&E) to assess the extent of inflammation or FluoroMyelin green (1:300; Thermo Fisher, Waltham, MA, USA), for specific labeling of myelin. For the detection of activated microglia and macrophages, slices were fixed with 4% paraformaldehyde for 10 min at room temperature (RT), washed three times with PBS and blocked for 4 h at RT with PBS containing 5% bovine serum albumin (BSA; Merck KGaA), 1% normal goat serum (NGS; Merck KGaA) and 0.3% Triton X-100 (Merck KGaA). Afterward, slices were incubated with the primary antibody (rabbit anti-mouse Iba-1; 1:1000, FUJIFILM Wako Pure Chemicals Corporation, Osaka, Japan) at 4 °C overnight. The primary antibody was diluted with PBS containing 5% BSA and 1% NGS. After washing, slices were covered with goat anti-rabbit Cy3 (1:400) for 1 h at RT. For testing autofluorescence of the samples as well as sensitivity and affinity of the used antibodies, negative controls were obtained, and no specific signal was detectable (not shown). Following final washing, sections were mounted with Fluoromount-G, including 4′,6-diamidino-2-phenylindole (DAPI; Thermo Fisher). Stainings were examined by microscopy (Axiophot2, Zeiss, Oberkochen, Germany) and analyzed in a blinded manner using Image J software (National Institute of Mental Health, Bethesda, ML, USA). To assess the degree of neuroinflammation, inflammatory and demyelinated areas were quantified and presented as the percentage of total area. For the examination of activated microglia and macrophages, Iba-1^+^ cells were counted and presented as the number of positive cells per mm^2^ of EAE lesion. For each animal, the arithmetic mean was calculated from five randomly selected coverslips with at least 3 slices per coverslip.

### 4.6. Immunological Assessment of EAE

For the extraction of CNS infiltrating cells, spinal cords were harvested after intensive transcardial perfusion with DPBS. CNS tissues were mechanically homogenized in DPBS, layered on a 30–50% Percoll (Merck KGaA) gradient and centrifuged for 30 min at 1200× *g* without using the brake. Calibrite beads (BD) were added before washing and staining for quantification of cell numbers isolated from the interphase. To assess the number of infiltrated platelets, CNS homogenates were centrifuged at 250× *g* for 5 min at RT to receive platelet-rich supernatant. This supernatant was stained for CD41 (clone MWReg30, BioLegend, San Diego, CA, USA), as well as CD61 (clone 2C9.G2, BioLegend), and fixed with 1% PFA before evaluation. CNS infiltrating cells were stained with fluorochrome-conjugated antibodies from BioLegend for CD3 (clone 17A2), CD4 (clone GK1.5), CD8a (clone 53-6.7), CD11b (clone M1/70), CD45R/B220 (clone RA3-6B2) and CD45 (clone 30-F11). Corresponding isotype controls were used for all stainings and for blocking the Fc receptor binding; cells were preincubated with an anti-CD16/CD32 antibody (BioLegend) for 5 min on ice. The stained samples were analyzed using a multi-color flow cytometer (Gallios, Beckman Coulter, Krefeld, Germany) and Kaluza software (Beckman Coulter). Cell doublets were excluded to ensure single cell counting. 

### 4.7. Quantification of Soluble Analytes

The retrobulbar murine blood was collected and centrifuged at 2200× *g* for 10 min to obtain blood plasma. Thromboxane B2 (Elabscience, Houston, TX, USA), PF4 (Merck KGaA) and GPVI (LSBio, Seattle, WA, USA) were quantified using enzyme-linked immunosorbent assays in blood plasma samples following the manufacturer’s recommendations.

### 4.8. Statistical Analysis

For each type of experiment, group sizes are given in the figure legends and data are presented as the mean ± standard error of the mean. No data outliers were excluded. For comparison of EAE scores between groups, a two-way analysis of variance (ANOVA) was performed, followed by a Bonferroni post-hoc test. D’Agostino-Pearson omnibus was used to assess the normality of a dataset. In the case of multiple comparisons, Kruskal–Wallis test with Dunn post hoc analysis was used. Otherwise, pairwise comparisons between groups were conducted using the Mann–Whitney U-test. Data were analyzed using Prism 5.04 (GraphPad Software, San Diego, CA, USA), and values of probability *p* < 0.05 were considered statistically significant. The level of significance was labeled as NS (not significant), * *p* < 0.05, ** *p* < 0.01 or *** *p* < 0.001.

## Figures and Tables

**Figure 1 ijms-22-09915-f001:**
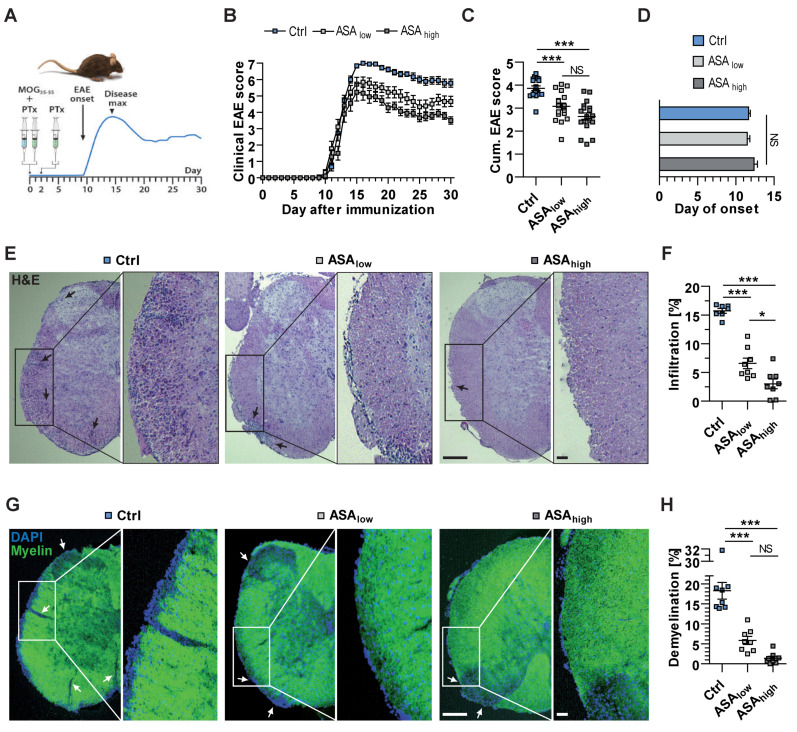
Acetylsalicylic acid (ASA) reduces the clinical disease activity of experimental autoimmune encephalomyelitis (EAE). (**A**) Experimental overview of active EAE induction illustrating the time points of myelin oligodendrocyte glycoprotein (MOG)_35–55_ peptide and pertussis toxin (PTx) administration. (**B**) The clinical EAE scores and (**C**) the cumulative (cum.) scores of MOG_35–55_-immunized mice are presented for mice receiving a vehicle as control (Ctrl), 3 mg/kg body weight (BW) ASA (ASA_low_) or 30 mg/kg BW ASA (ASA_high_) daily, beginning on the day of immunization. (**D**) The average day of disease onset is presented for each experimental group. (**E**–**H**) Lumbar spinal cord sections isolated at disease maximum (day 16) were histologically analyzed using hematoxylin and eosin (**H**,**E**), (**E**,**F**) as well as myelin staining (Myelin) (**G**,**H**). For the visualization of cell nuclei, sections were stained with 4′,6-diamidino-2-phenylindole (DAPI), of which representative images are shown (scale bar: 200 µm; close-up scale bar: 100 µm). Arrows indicate areas with extensive infiltration (**E**) or demyelination (**G**) and quantitative analyses are shown (**F**,**H**). Data were analyzed by two-way analysis of variance (ANOVA; 18 vs. 18 vs. 18 mice; 3 independent experiments; (**B**)) or Mann–Whitney U-test (**C**,**D**,**F**,**H**). Each symbol represents an individual mouse (**C**,**F**,**H**). The level of significance was labeled as NS (not significant), * probability *p* < 0.05 or *** *p* < 0.001.

**Figure 2 ijms-22-09915-f002:**
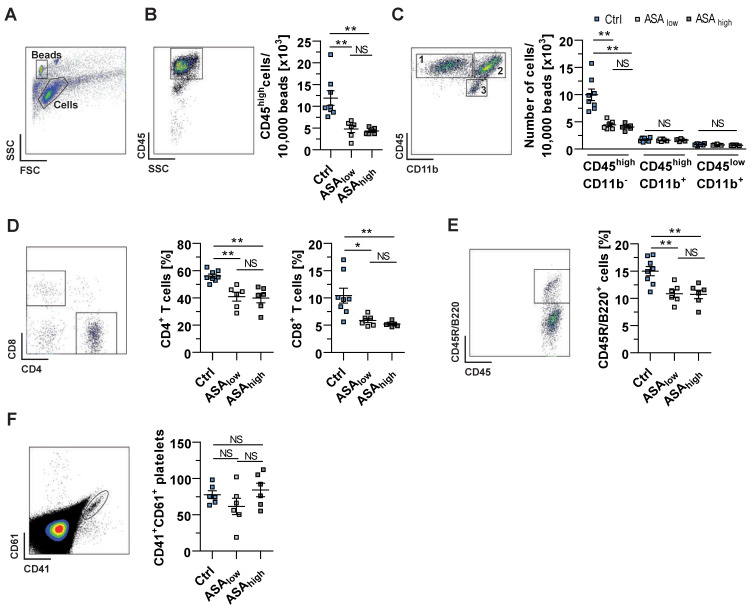
Central nervous system (CNS) immune cell infiltration is reduced by low-dose ASA administration. (**A**) CNS-infiltrating immune cells of all experimental groups were analyzed at disease maximum using flow cytometry. (**B**) The number of clusters of differentiation CD45^high^ cells, representing infiltrated peripheral immune cells, was quantified in relation to reference beads. (**C**) For further insights, infiltrated cells were analyzed for CD45^high^CD11b^−^ (1, indicating peripheral lymphoid cells), CD45^high^CD11b^+^ (2, indicating peripheral myeloid cells) and CD45^low^CD11b^+^ cells (3, indicating resident microglial cells) relative to reference beads. (**D**,**E**) For a broader characterization of infiltrating lymphoid cells, the proportions of CD4^+^CD3^+^, CD8^+^ CD3^+^ and CD45R/B220^+^ cells were determined. (**F**) Platelet-rich supernatants isolated from CNS homogenates were stained for CD41 and CD61 and examined by flow cytometry for quantification of platelets present in the CNS. A representative dot plot is shown, and total numbers of CD41^+^CD61^+^ cells were calculated. Data were analyzed by Mann–Whitney U-test (**B**–**F**), and each symbol represents an individual mouse (8 vs. 6 vs. 6 mice; 2 independent experiments; **B**–**F**). The level of significance was labeled as NS, * *p* < 0.05 or ** *p* < 0.01. FSC, forward scatter; SSC, side scatter.

**Figure 3 ijms-22-09915-f003:**
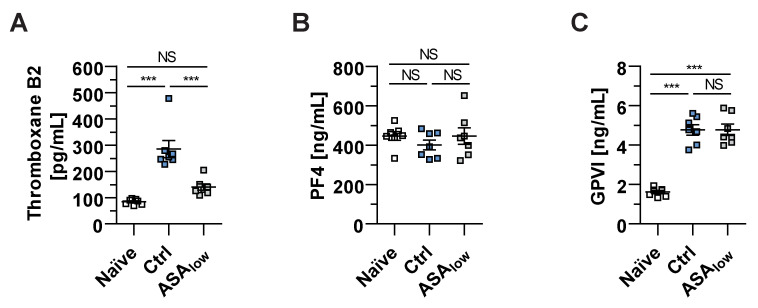
Administration of low-dose ASA reduces the synthesis of thromboxane B2 during neuroinflammation. (**A**–**C**) The levels of thromboxane B2 (**A**), platelet factor 4 (PF4) (**B**) and glycoprotein VI (GPVI) (**C**) were determined using enzyme-linked immunosorbent assays. Therefore, blood plasma was obtained from naïve mice (Naïve), from vehicle-treated mice at EAE disease maximum (Ctrl) and mice treated with 3 mg/kg BW of ASA (ASA_low_) at EAE disease maximum. Data were analyzed by Mann–Whitney U-test (7 vs. 7 vs. 7 mice; 2 independent experiments). Each symbol represents an individual animal. The level of significance was labeled as NS, *** *p* < 0.001.

**Figure 4 ijms-22-09915-f004:**
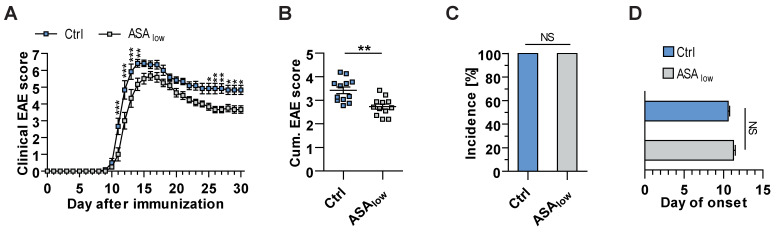
Therapeutic administration of low-dose ASA ameliorates clinical symptoms of EAE. (**A**,**B**) Daily clinical EAE scores of MOG_35–55_-immunized mice are presented (**A**) and the cum. EAE scores are calculated for vehicle (Ctrl) and low-dose ASA-treated mice (ASA_low_) (**B**). Mice were treated with ASA or vehicle from the day of clinical EAE onset. (**C**,**D**) Disease incidence (**C**) and the average day of disease onset after immunization (**D**) are shown. Data were analyzed by two-way ANOVA (*n* = 12 vs. 12 mice; two independent experiments; (**A**)) or Mann–Whitney U-test (**B**–**D**). Each symbol represents an individual mouse (**B**). The level of significance was labeled as NS, * *p* < 0.05, ** *p* < 0.01 or *** *p* < 0.001.

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
