# Peer review of "Platelet Inhibition by Low-Dose Acetylsalicylic Acid Reduces Neuroinflammation in an Animal Model of Multiple Sclerosis"

_ijms, 2021, doi:10.3390/ijms22189915_

Round 1

Reviewer 1 Report

In this manuscript, the authors found that low-dose ASA could reduce the clinical disease activity of EAE, CNS-infiltrated CD4+ T cells, and TxA2 synthesis. ASA could also improve the clinical outcome of EAE. This study has significance for providing a potential treatment approach for MS.

Some comments:

  1. The title of Figure 1 would be better to be changed, because this figure contains three groups (ctrl, low, and high), and it’s not appropriate to mention only low ASA effect. In addition, the scale bar of HE is not clearly indicated, and the significance marks do not correspond to Figure 1F (*P).
  2. Line 177, PF4 requires the full name to be provided in the main text.
  3. In the discussion section, it would be desirable for the authors to conduct a discussion based on the experimental results.
  4. It would be advisable for the authors to clarify the relationship between thromboxane A2 and B2.
  5. What is the difference between Figure 1A-D and Figure 4A-D in terms of experimental design, method, and purpose? They seem to be identical.
  6. It’s advisable for the authors to briefly describe the significant findings of this manuscript in the summary section.

Reviewer 2 Report

In this article, Vogelsang et al focus on the platelet-mediated impact on neuroinflammation following ASA treatment in an animal model of MS.

The main aim of this study was to demonstrate that an antiplatelet agent (acetylsalicyclic acid or ASA), could inhibit neuroinflammation by affecting platelets if applied at low-dose using experimental autoimmune encephalomyelitis (EAE), as a model for MS. The authors found that oral administration of low-dose ASA alleviated symptoms of EAE and this was accompanied by reduced inflammatory infiltrates and less extensive demyelination. They demonstrated a 15% reduction of CNS-infiltrated CD4+ T cells in ASA-treated mice compared to control animals at the height of the disease when using flow cytometry as the analysis tool. The authors state that the concentrations of platelet factor 4 and glycoprotein VI (GPVI) were not affected by low-dose ASA treatment whereas the plasma levels of thromboxane Awere decreased. Under the experimental conditions used in this study, the authors suggest that low-dose ASA could ameliorate the platelet-dependent neuroinflammatory response in vivo, and in turn this could lead on to a potential treatment approach for MS in the future.

Throughout this article, Vogelsang et al give a good explanation of other publications in this field of research and clearly highlight the similarities and differences between their work and previously published data (both positive and negative). This is nice to see as it does give a good balance of views and expectations in this area.

The authors state that targeting GPVI is a less compelling treatment strategy for autoimmune neuroinflammation as it is a less effective abrogator of the neuroinflammatory response. GPVI acts as a platelet activation receptor and so its function may well be compensated for by alternative pathways. Comparative studies have demonstrated that a combined approach including both GPVI inhibition and ASA treatment can lead to an additional decrease of plaque-induced platelet aggregation in vitro. Based on this, the authors suggest that a combination therapy of GPVI blockade and ASA could be a promising therapeutic approach to assess in EAE.

The authors conclude that their data suggest that the inhibition of platelet activation via low-dose ASA is beneficial in an animal model of MS. They comment that their findings indicate a potential antithrombotic strategy to combat the effects of MS.

Main points and comments:

  1. This article is reasonably well-written, is fairly easy to read and it does present some nice clear-cut data with some sensible suggestions and reasoning based on the results. The data support the suggestions from the authors and the conclusions are fairly well substantiated throughout the manuscript. The assays that have been employed by the authors are suitable for this type of analysis and the materials and methods are logical.
  2. Lines 44 to 60 make a lot of statements without any references. Can the authors please rectify this?
  3. The authors highlight the efficiency of platelet depletion with regards to the reduction in disease severity during the effector phase of EAE, along with a reduction in CNS-infiltrating immune cells from the periphery. They discuss the fact that platelets are chronically activated in relapsing-remitting and secondary progressive MS and they highlight the effects of both low-dose and high-dose ASA and the various implications associated with the use of ASA in the clinical setting.
  4. The authors state that there has not been a lot of research carried out on assessing the effect of ASA in EAE disease progression. Can the authors elaborate on why this may be the case? Why has this area of research been left behind in terms of experimental research?
  5. Being really picky, the arrow on Figure 1A showing maximum disease looks as though it is pointing to day 14 and not day 16 as is mentioned in the text for the IHC. Can the authors please adjust this to give some clarity?
  6. Figure 2 certainly demonstrates the low number of CD4+ T cells, CD8+ T cells and B cells following low-dose or high-dose ASA treatment. Can the authors suggest why there was very little difference between the low-dose and high-dose ASA results as shown in Figure 2 with regards to the cumulative EAE scores, the degree of demyelination, the total number of CNS infiltrating leukocytes, lymphocytes, CD4+ T cells, CD8+ T cells and B cells?
  7. In the Materials and Methods (lines 354 to 356), the authors state “….their last observed score was assumed for the rest of the experiment. The cumulative score was calculated as the average of all daily scores for the whole period of the experiment.” Can the authors please explain what this actually means as the 2 comments appear to contradict each other. If the last observed score was then assumed for the rest of the experiment….how can the authors then acquire a cumulative score that is calculated as the average of daily scores? Did they use daily scores or did they base everything on the last observed score? If the latter, where does the average score come into all this? I found this a little bit confusing and so I would request that the authors clarify this point in the Materials and Methods section to avoid any ambiguity.
  8. Lines 377 to 378 – The authors state that “Negative controls were obtained and no specific signal was detectable (not shown).” Can the authors please be more detailed with their explanation of their controls. The comment shown above seems to appear in the middle of nowhere and does not have any real description. I am confused as to what controls they are talking about. This comment does seem to be in the wrong place in the text. Other sections of the Materials and Methods give much better explanations of the control samples and techniques used. Please can the authors clarify their statement and move this sentence to the correct place. Also, I would personally like to see examples of the negative controls within the manuscript as they are just as important as positive controls from the point of view of comparison.
  9. The sentence structure needs some attention in a few places (such as lines 68, 76, 312 and 313.
  10. There are a few typographical issues with the odd extra “s” in a number of places (such as line 48, 54, 141).
  11. In this article, the results demonstrate that pharmacologic blockade of platelets using both low-dose and high-dose ASA renders EAE mice less susceptible to neuroinflammation. The authors have very nicely shown that there are meaningful relationships between low-dose ASA-induced platelet inhibition and amelioration of disease pathogenesis. They have also highlighted the point that low-dose ASA has just as good an anti-inflammatory effect on neuroinflammation as the high-dose ASA but the low-dose ASA has less side effects. The results generated in this article suggest that the beneficial effect of ASA treatment during EAE is mediated by altered platelet function.
  12. The authors found an interesting difference between naïve and EAE mice in that soluble GPVI plasma levels were increased during EAE disease progression compared to naïve mice. These data suggest an important role of GPVI for platelets during neuroinflamnmation.
  13. I am very pleased to see that no data outliers have been excluded from the data sets.
  14. Just a small point. On my version of the Figures, Figure 2 was very difficult to tell the difference between ASA high and low from the symbols on the actual Figure. I think this may just be the output at this end but can this be checked to make sure the colours do show up clearly for the reader?
